# Combating Black Fungus: Using Allicin as a Potent Antifungal Agent against Mucorales

**DOI:** 10.3390/ijms242417519

**Published:** 2023-12-15

**Authors:** Christina Schier, Martin C. H. Gruhlke, Georg Reucher, Alan J. Slusarenko, Lothar Rink

**Affiliations:** 1Department of Plant Physiology, RWTH Aachen University, Worringer Weg 1, 52074 Aachen, Germany; christina.schier@rwth-aachen.de (C.S.); alan.slusarenko@bio3.rwth-aachen.de (A.J.S.); 2Institute of Immunology, RWTH Aachen University Hospital, Pauwelsstraße 30, 52074 Aachen, Germany; georg.harald.reucher@rwth-aachen.de; 3GENAWIF e.V.—Society for Natural Compound and Active Ingredient Research, Lukasstraße 1, 52070 Aachen, Germany; martin.gruhlke@genawif.com; 4Institute of Applied Microbiology—iAMB, Aachener Biology and Biotechnology—ABBt, RWTH Aachen University, 52074 Aachen, Germany

**Keywords:** allicin, amphotericin B, antimycotic, COVID-19, glutathione reductase, mucormycosis

## Abstract

Invasive fungal (IF) diseases are a leading global cause of mortality, particularly among immunocompromised individuals. The SARS-CoV-2 pandemic further exacerbated this scenario, intensifying comorbid IF infections such as mucormycoses of the nasopharynx. In the work reported here, it is shown that zygomycetes, significant contributors to mycoses, are sensitive to the natural product allicin. Inhibition of Mucorales fungi by allicin in solution and by allicin vapor was demonstrated. Mathematical modeling showed that the efficacy of allicin vapor is comparable to direct contact with the commercially available antifungal agent amphotericin B (ampB). Furthermore, the study revealed a synergistic interaction between allicin and the non-volatile ampB. The toxicity of allicin solution to human cell lines was evaluated and it was found that the half maximal effective concentration (EC_50_) of allicin was 25–72 times higher in the cell lines as compared to the fungal spores. Fungal allicin sensitivity depends on the spore concentration, as demonstrated in a drop test. This study shows the potential of allicin, a sulfur-containing defense compound from garlic, to combat zygomycete fungi. The findings underscore allicin’s promise for applications in infections of the nasopharynx via inhalation, suggesting a novel therapeutic avenue against challenging fungal infections.

## 1. Introduction

Globally, infectious diseases stand as primary drivers of mortality. In 2019, an estimated 1.2 million people died from antibiotic-resistant bacterial infections and about 4.95 million more deaths were associated with antimicrobial resistance [1]. In addition to this major global health threat, invasive fungal (IF) diseases are emerging with increased frequency, especially in immunocompromised populations. The incidence of fungal infections is largely derived from estimates, as many fungal diseases remain undiagnosed, and there is limited available data on vulnerable populations and disease prevalence [2].

During the SARS-CoV-2 pandemic, an increase in the incidence of comorbid IF infections was observed, particularly aspergillosis, mucormycosis and candidiasis [3]. The situation has aggravated to such an extent that the World Health Organization (WHO) published a “fungal priority pathogens list” in October 2022, ranking 19 fungal pathogens in terms of public health importance based on ten criteria [4]. The criteria included deaths, annual incidence, and resistance to antifungals. The latter poses a major problem: antifungal resistance is on the rise [5,6] and for clinical use, only four classes of antifungal drugs (polyenes, azoles, echinocandins and pyrimidines) are used for the treatment of fungal diseases, while the development of new drugs is progressing slowly [7,8].

Mucormycosis, which is popularly known as “black fungus”, is caused by fungi of the order Mucorales. The most common organisms isolated from mucormycosis patients are *Rhizopus* species, but also species of the genera *Mucor*, *Lichtheimia* and *Apophysomyces* are common causative agents [9,10]. Infection occurs after spore inhalation, which is why it primarily affects sinuses and the lungs. Furthermore, it can spread to the eye, the central nervous system and the gastrointestinal tract. With the emergence of COVID-19, the number of mucormycosis infections increased drastically, particularly in India and other low-income and middle-income countries [11,12]. Common risk factors for mucormycosis are a weakened immune system, e.g., due to illness or medication, diabetes mellitus and, as in COVID-19 associated mucormycosis (CAM), primary infections [13]. Treatment recommendations for adults were published by the European Confederation of Medical Mycology (ECMM) and the Mycoses Study Group Education & Research Consortium (MSG ERC) in 2019 [14]. They include first-line antifungal monotherapy with lipid formulations of amphotericin B (ampB), isavuconazole and posaconazole given intravenously or orally. Combination therapy with ampB and an azole antifungal is also recommended as well as the surgical removal of infected tissues. A metanalytic study from 2021, which analyzed 79 studies, found that 854 of 1544 subjects with pulmonary mucormycosis died, giving a pooled mortality of 57.1% [15]. The reported mortality ranged from 51.7% to 62.6% (I^2^ = 83.5%) in the individual studies. The combination of drug and surgical therapy achieved significantly better survival rates than drug treatment alone. However, treating mucormycosis can be a financial challenge, especially for low-income countries, but also the US, due to the high cost of drugs, surgery and hospitalization [16].

Because of rising numbers of mucormycosis infections in low- and middle-income countries as well as the increased occurrence of antifungal resistance, new and affordable therapeutic approaches are desirable. In silico molecular docking analyses identified inhibitory ligands for specific virulence proteins of *Rhizopus delemar* [17]. In addition to the triazole antifungal agents pramiconazole and saperconazole, some natural bioactive compounds were identified as potential ligands against Mucorales, including vialinin B, a dibenzofuran compound isolated from the *Thelephora vialis* mushroom, metabolites from sponges (12,28-oxamanzamine A and deoxytopsentin) or hesperidin, a flavanone glycoside from citrus fruits. Furthermore, the inhibitory effect of essential oil from *Eucalyptus polybrachtea*, crude aqueous omam extract, and garlic extract were shown in vitro [18,19].

A potential therapeutic candidate could be allicin (diallyl thiosulfinate) which is a sulfur-containing defense substance from garlic (*Allium sativum*) first described and investigated in 1944 [20]. Allicin is naturally formed after plant tissue injury. The precursor molecule alliin (*S*-allyl-L-cysteine sulfoxide) is enzymatically cleaved and releases allyl sulfenic acid. Subsequently, allicin is formed by spontaneous condensation of two allyl sulfenic acid molecules. As a thiol trapping reagent, it has a broad spectrum of antimicrobial and biocidal functions, for example against phyto- and human-pathogenic bacteria [21,22], including multi drug resistant lung pathogens, as well as fungi [23,24] and parasites [25,26]. Virucidal effects of allicin have also been described [27,28]. Interestingly, allicin is not only active in solution, but also as a vapour [21,23]. This results in a great potential for applications utilizing inhalation, for example in the case of nasal Mucorales infection.

The mode of action of allicin is based on the reversible oxidation of both protein thiols and low-molecular weight thiols such as glutathione, producing *S*-thioallylated derivatives [29], which, depending upon concentration, can result in oxidative stress. The impact of *S*-thioallylation has been studied in both human cells and pathogenic microorganisms [29,30]. It can affect the function of structural proteins or even lead to the inactivation of important enzymes and thus to the impairment of cell signaling pathways [31,32]. The cytoskeleton is also a target of allicin [29,33]. Furthermore, the immune system undergoes modulation through allicin stimulation, a factor that gains importance, such as in the context of a COVID-19 infection [34].

This study aimed to demonstrate the sensitivity of zygomycetes, a significant contributor to mycoses, to the natural substance allicin. Through direct contact and gas phase exposure, the research established this sensitivity and explored the synergistic potential of allicin with the antifungal agent ampB. Furthermore, the study aimed to assess the potential for reducing the required concentration of ampB, thereby contributing to more effective infection containment, reduced antifungal side effects and improved patient protection. The cytotoxicity of allicin to human epithelial cell lines was investigated in order to make an initial risk assessment regarding the potential therapeutic use of allicin.

## 2. Results

### 2.1. Morphological Characterisation of Mucorales Species

Growth of commercially acquired fungi *Rhizopus stolonifer* and *Mucor racemosus* was studied on different culture media. Czapek agar (CZA), malt extract agar (MEA), potato dextrose agar (PDA), Sabouraud glucose agar (SGA) and oatmeal agar (OA) were used. After 40 h, plates inoculated with *R. stolonifer* were completely covered in PDA, SGA and OA media, resulting in no clear distinction between colonies (Figure 1A). In addition, aerial mycelium with sporadic terminal pigmented sporangia was macroscopically visible. Microscopically, reproductive structures of *R. stolonifer* could be identified, consisting of rhizoids, stolon, sporangiophore and a terminal sporangium in which endogenous sporangiospores are formed.

*M. racemosus* showed less vigorous growth after 40 h incubation on all culture media compared to *R. stolonifer.* Colonies were clearly demarcated from each other on all culture media (Figure 1B). Hyphae of *M. racemosus* were highly branched and single reproductive structures could be observed.

### 2.2. Allicin Susceptibility Depends on Spore Amount

To quantify allicin’s activity on the Mucorales, allicin was incorporated into molten agar at different concentrations and poured into Petri plates. Double distilled water (H_2_O_dd_) was used as a control. A total of 20 µL of a serial dilution of fungal spores was spotted onto the medium containing allicin. Growth was photographically documented after incubation at 22 °C for 24 h. Both *R. stolonifer* (Figure 2A) and *M. racemosus* (Figure 2B) showed allicin sensitivity. At 20 µM allicin, only spores at the two highest spore densities grew and 40 µM allicin completely inhibited spore germination. In general, *M. racemosus* showed less growth than *R. stolonifer* and was also somewhat more susceptible to allicin.

### 2.3. Allicin Inhibits Mucorales Spore Germination by Diffusion and as a Vapour

Allicin susceptibility was studied in an agar diffusion test (Figure 3A). A total of 40 µL of aqueous allicin solution (10 mM or 20 mM) was used. Triplicates were made and double distilled water was used as a control. Inhibition zone diameters were measured and the mean inhibition zone area was calculated (Figure 3B). Using 10 mM allicin solution, a mean area of inhibition of 890 mm^2^ (*SD* = 112) formed in the case of *R. stolonifer*. The mean inhibition zone area of *M. racemosus* was 912 mm^2^ (*SD* = 139). In the presence of 20 mM allicin, mean inhibition zone area of *R. stolonifer* was 1154 mm^2^ (*SD* = 124) and that of *M. racemosus* was 1386 mm^2^ (*SD* = 66).

An inverted inhibition zone test, where the drop of allicin solution is placed on the lid and the seeded agar inverted over it, showed that allicin also inhibits fungal growth via the gas phase (Figure 3C). The test was performed in triplicate. Inhibition zone diameters were measured and the mean inhibition zone area was calculated (Figure 3D). Since growth of *M. racemosus* could not be observed after 24 h, pictures were taken after 48 h. All those plates with an inhibition zone showed a zone of reduced fungal growth around the growth-free inhibition zone. No growth inhibition was observed on plates treated with H_2_O_dd_ as well as 96% ethanol. When treated with 25 µL of 25 mM allicin solution, the average inhibition zone area occurring for *R. stolonifer* was 1453 mm^2^ (*SD* = 68), *M. racemosus* had an average inhibition zone area of 1215 mm^2^ (*SD* = 36) with the same treatment. Doubling the allicin concentration resulted an average inhibition zone area of 1421 mm^2^ (*SD* = 349) in case of *R. stolonifer* and 2018 mm^2^ (*SD* = 167) for *M. racemosus*. Application of 50 µL 50 mM allicin resulted the largest inhibition zones for both fungi. Here, the mean inhibition zone area of *R. stolonifer* was 1965 mm^2^ (*SD* = 137) and 2405 mm^2^ (*SD* = 100) for *M. racemosus*.

### 2.4. Allicin Inhibits Spore Germination More Effectively Than Amphotericin B upon Direct Contact

Allicin effectiveness was then compared with that of amphotericin B (ampB), a commercially available antifungal agent (Figure 4A). The experiment was performed three times. The half maximal effective concentration (EC_50_; Figure 4B) and the minimum inhibitory concentration (MIC) were determined. Regarding the MIC, it is evident that allicin has an overall lower MIC value compared to ampB, which could not be determined exactly with the concentrations used (MIC > 250 µM). Furthermore, *Mucor racemosus* had a higher MIC for allicin (62.5 µM) than *Rhizopus stolonifer*, whose MIC was 31.3 µM.

In terms of the EC_50_, lower values were also found with allicin in general and *R. stolonifer* showed significantly higher susceptibility to both antifungals compared to *M. racemosus*. For *R. stolonifer*, 3.2 µM allicin solution was sufficient to inhibit 50% of spore germination, and for *M. racemosus*, 9.2 µM allicin was sufficient. In comparison, ampB treatment required concentrations approximately 34 times higher than allicin to inhibit 50% of *Rhizopus* spore germination (EC_50_ = 108 µM). For Mucor, the EC_50_ for ampB was 206 µM, about 22 times higher than for allicin.

### 2.5. The Efficacy of Allicin as a Vapour in Inhibiting Spore Germination Is Comparable to That of a Direct Application of Amphotericin B

As previously shown, allicin inhibits spore germination more effectively than ampB upon direct contact (Figure 4). However, since the direct use of allicin is not recommended due to cytotoxic properties [35,36,37], treatment with allicin vapour is considered to have greater therapeutic applicability. Therefore, the concentrations at which inhibition of spore germination occurs for allicin application via vapour, diffusion and direct contact were compared to conclude whether vapor application is viable in comparison to ampB direct application. Based on the inverted inhibition zone tests (Figure 3C,D), the actual effective allicin concentration leading to inhibition of spore germination in *R. stolonifer* and *M. racemosus* was determined. In addition, the effective concentration in the agar diffusion assay was estimated to allow comparison of the different susceptibility assays. The calculations were based on the highest substance amount of allicin used in the experiments.

In order to determine the effective allicin concentration (c_iz_) within the inhibition zone, the amount of allicin deposited within the inhibition zone’s volume had to be estimated. However, a number of variables were unknown: after complete evaporation of the droplet over 24 h, there is an equilibrium between allicin in the agar and in the gas phase as the system has achieved high entropy. Since this ratio is unknown, it was assumed that there was no allicin left in the gas phase as it completely diffused into the agar, thus overestimating the total deposited allicin quantity. Secondly, the distribution of allicin throughout the agar had to be modeled. Leontiev et al. found that allicin will evaporate evenly across the entire droplet surface [38]. This causes a nearly circular spread of allicin molecules in the gas phase around the droplet, yielding a higher allicin deposition in the agar center as this area is closest to the droplet and thus is exposed to allicin for a longer period of time. Accordingly, the allicin concentration will be maximal directly above the droplet and decreasing towards the outer agar boundary (Figure 5). Since the real allicin distribution is unknown, a square distribution was estimated by assuming the entire allicin to be evenly distributed inside the inhibition zone. This underestimated the peak concentration in the agar center. However, only the concentration at the inhibition zone’s boundary was of interest, which was again overestimated. For a better classification of the results, calculations were also made for the boundary cases in which all the allicin was equally deposited across the entire agar (Figure 5, left) or only deposited directly above the droplet respectively (Figure 5, right). Finally, it was assumed that the allicin in the inhibition zone only interacted with the embedded spores whereas in reality, parts of the allicin would be buffered due to the agar’s components. Likewise, this assumption lead to an overestimation of the actual effective allicin concentration.

First, a lower limit c_lo_ (Figure 5, left) for the effective allicin concentration in the inverted inhibition zone test was approximated. A homogeneous distribution of allicin molecules over the entire agar surface A_max_ in the Petri dish (r_max_ = 45 mm) was assumed. To ensure comparability with the agar diffusion assay and direct application, the allicin was estimated to be evenly distributed throughout the agar. By considering the thickness of the agar layer h = 5 mm, the area concentration was converted to a volume concentration. The maximum amount of substance of allicin (n_max_) used in the experiment was calculated using the applied allicin concentration c = 50 mM and the volume V = 50 µL. Inserting the values in Formula (1) yielded the lower limit of the allicin concentration as vapor
(1)$$c_{\mathrm{lo}}=\frac{n_{\max}}{A_{max}\times \;h}=\frac{c\;\times \;V}{\pi {\left(r_{\max}\right)}^{2}\;\times \;h}=\frac{50\;\times \;10^{-3}\frac{\mathrm{mol}}{L}\;\times \;50\;\times \;10^{-6}\;L}{\pi \;\times \;({45\;\mathrm{mm})}^{2}\;\times \;5\;\mathrm{mm}}\;\approx \;78.6\;\frac{\mathrm{pmol}}{{\mathrm{mm}}^{3}}=78.6\;\mu M.$$

Next, an upper limit c_up_ for the effective allicin concentration was estimated. It was assumed that the allicin evaporates strictly vertically from the droplet and only enters the agar directly above the droplet. For this purpose, the radius of the 50 µL droplet r_min_ was calculated assuming a hemispherical droplet shape. From the equation for the hemispherical volume
(2)$$V_{\mathrm{hem}}=\frac{2}{3}\pi {r_{\min}}^{3}$$
followed
(3)$$r_{\min}=\sqrt[{3}]{\frac{3V_{\mathrm{hem}}}{2\pi }}\;\approx \;2.88\;\mathrm{mm}.$$

Again, the maximum amount of substance of allicin (n_max_) used in the experiment was calculated using the applied allicin concentration c = 50 mM and the volume V = 50 µL. With consideration of the agar layer thickness h = 5 mm, the upper limit of the effective allicin concentration was
(4)$$c_{\mathrm{up}}=\frac{n_{\max}}{A_{\min}\times \;h}=\frac{c\;\times \;V}{\pi {\left(r_{\min}\right)}^{2}\;\times \;h}=\frac{50\;\times \;10^{-3}\frac{\mathrm{mol}}{L}\;\times \;50\;\times \;10^{-6}\;L}{\pi \;\times \;({2.88\;\mathrm{mm})}^{2}\;\times \;5\;\mathrm{mm}}\;\approx \;19188\;\frac{\mathrm{pmol}}{{\mathrm{mm}}^{3}}=19188\;\mu M.$$

The mean effective allicin concentration $\bar{c}$_iz_ (Figure 5, center) was calculated in relation to the mean inhibition zone area Ā_iz_ for each Mucorales species. Here, it was assumed that all the allicin spread homogeneously only in the agar of the inhibition zone area where spore germination was fully inhibited. This allowed approximation of a mean effective allicin concentration leading to inhibition of spore germination at the border of the inhibition zone. Inserting the corresponding parameters into the equation
(5)$${\bar{c}}_{\mathrm{iz}}=\frac{n_{\max}}{{\bar{A}}_{\mathrm{iz}}\;\times \;h}=\frac{c\;\times \;V}{\pi ({{\bar{r}}_{\mathrm{iz}})}^{2}\;\times \;h}=\frac{50\;\times \;10^{-3}\frac{\mathrm{mol}}{L}\;\times \;50\;\times \;10^{-6}\;L}{\pi \;\times \;({{\bar{r}}_{\mathrm{iz}})}^{2}\;\times \;5\;\mathrm{mm}}$$
yielded a mean effective allicin concentration of 255 µM for *R. stolonifer* (${\bar{r}}_{\mathrm{iz}}=25\;\mathrm{mm})$ and 210 µM for *M. racemosus* (${\bar{r}}_{\mathrm{iz}}=27.5\;\mathrm{mm})$. This resulted in the final inequality
c_lo_ = 78.6 µM < 210 µM < 255 µM < 19188 µM = c_up_,(6)
which provided an indication of the actual effective allicin concentration against Mucorales via vapor.

The abovementioned calculations to determine the mean effective allicin concentration ${\bar{c}}_{\mathrm{iz}}$ were also carried out for the agar diffusion experiments (Section 2.3), assuming the allicin to be confined to the inhibition zone and thus overestimating the actual concentration. ${\bar{c}}_{\mathrm{iz}}$ resulted from the quotient of the maximum amount of allicin used (n_max_) and the total inhibition area (Ā_total_) multiplied by the agar layer thickness h = 5 mm. n_max_ was obtained by multiplying the applied allicin concentration c = 20 mM with the volume V = 40 µL. For the calculation of Ā_total_, the area of the application well (r_well_ = 3 mm) was subtracted from the total inhibition zone area, resulting in the final equation
(7)$${\bar{c}}_{\mathrm{iz}}=\frac{n_{\max}}{{\bar{A}}_{\mathrm{total}}\;\times \;h}=\frac{c\;\times \;V}{\pi \;(({{\bar{r}}_{\mathrm{iz}})}^{2}\;-\;({r_{\mathrm{well}})}^{2})\;\times \;h}.$$

Inserting the described parameters and the average radius of the inhibition zone resulted in a mean effective allicin concentration for *R. stolonifer* (${\bar{r}}_{\mathrm{iz}}=19.2\;\mathrm{mm})$ of 142 µM and a mean effective allicin concentration for *M. racemosus* (${\bar{r}}_{\mathrm{iz}}=21\;\mathrm{mm})$ of 118 µM.

Finally, the effective concentrations of allicin and ampB via direct contact, which were shown by the minimum inhibitory concentration (MIC; Section 2.4), were added to the consideration to enable a comparison of ampB application with allicin gas phase application. The MIC for allicin was 31.3 µM for *R. stolonifer* and 31.3 µM for *M. racemosus*, while the MIC of ampB was >250 µM for both fungi.

The calculated and experimental effective concentrations of the different application forms are summarized in Table 1.

The comparison of the different concentrations led to the following conclusions:*R. stolonifer* seems to be less susceptible to allicin than *M. racemosus* for vapor and diffusion treatment.Allicin application via diffusion is approx. 1.8 times more effective than vapor regardless of the species.Allicin vapor reaches effective concentrations of the same order of magnitude as ampBs’ direct application. Thus, inhibition of spore germination is more or equally effective with allicin vapor.

Given the crude overestimations for both vapor and diffusion experiments, the actual mean effective concentrations of those experiments are most certainly below the values given in Table 1. Thus, any form of allicin application can be considered more effective than a direct ampB application.

### 2.6. Correlation between Allicin Resistance, Glutathione Reductase Activity and Cellular Glutathione Levels

To investigate the connection between allicin resistance and glutathione reductase (GR) activity in the Mucorales species, enzyme activity was determined photometrically with dithionitrobenzoic acid (DTNB). Using the amount of protein in the cell lysate determined by Bradford assay, GR activity per mg protein was calculated (Figure 6A). In addition, cellular glutathione content of the lysates was determined and the ratio of oxidized glutathione disulfide (GSSG) to reduced glutathione (GSH) was determined (Figure 6B). Overall, GR activity and glutathione levels were significantly lower in *M. racemosus* than in *R. stolonifer*. GR activity was 99 nkat per mg protein (*SD* = 15) and thus about 1.8 times lower than in *R. stolonifer*, which was 176 nkat per mg protein (*SD* = 27). Total glutathione content was also significantly higher in *R. stolonifer* with 633 µM (*SD* = 118) compared to 135 µM (*SD* = 61) in *M. racemosus*. In both fungi, about 80% of the glutathione pool was in the reduced GSH form.

### 2.7. Allicin and Amphotericin B Show a Synergistic Effect against Mucorales

It was investigated whether the synergistic effect of allicin with ampB reported in other fungi could also be demonstrated for the Mucorales [39,40,41]. Therefore, spore germination was studied upon treatment with combinations of allicin and ampB at different concentrations and the ratio of germinated spores to total spore number was determined (Figure 7). Again, fungal spore germination of *R. stolonifer* was evaluated after 24 h and that of *M. racemosus* after 48 h. Regarding *R. stolonifer*, 125 µM of ampB alone reduced spore germination to 31.2%; at the lower concentrations used, all spores germinated. The other ampB concentrations tested did not cause inhibition. Allicin alone inhibited spore germination up to a concentration of 3.9 µM; lower concentrations resulted in no significant inhibition. In the case of *M. racemosus*, at the tested concentrations ampB showed no inhibitory effect; and the same was true for allicin concentrations below 7.8 µM. For both fungi, the combination of both antifungals resulted in substantially lower concentrations of each antifungal being required to inhibit spore germination. Interestingly, the antifungal combination had a stronger effect on *M. racemosus*, as even the lowest concentrations of both antifungals had an inhibitory effect.

### 2.8. Allicin Has a Cytotoxic Effect on Human Epithelial Cell Lines

Lastly, it was examined whether allicin has cytotoxic properties in addition to its antifungal activity. The non-small cell lung cancer cell line A549 from human adenocarcinoma alveolar epithelial cells and the WISH cell line, which originated from a human papillomavirus-related cervical adenocarcinoma, were used for this purpose. Cells were incubated with allicin for 1 h and cell viability depending on the allicin concentration was assessed using colorimetric 3-(4,5-dimethylthiazol-2-yl)-2,5-diphenyltetrazolium bromide (MTT) assay (Figure 8). The experiment was performed three times with A549 and four times with WISH. Half maximal effective concentration (EC_50_) of allicin was calculated for both cell lines, which was 236 µM for A549 cells and 231 µM for WISH cells.

## 3. Discussion

Since the vast majority of fungi so far tested are quite sensitive to allicin [21,23,42], it was expected that zygomycetes would also show allicin sensitivity. However, since this class of fungi has so far not been extensively investigated with respect to allicin sensitivity, a simple drop test was first used to investigate the extent to which the two species tested are sensitive to allicin (Figure 2).

The growth of *Rhizopus stolonifer* on PDA medium without allicin was clearly more prolific than that of *Mucor racemosus*, so that this must also be taken into account for the subsequent consideration of allicin susceptibility. At a concentration of 40 µM in the medium, both strains were completely inhibited in growth; at lower concentrations, a concentration-dependent decrease in growth was seen, although *R. stolonifer* appeared less susceptible to allicin than *M. racemosus.*

Since allicin is volatile, we addressed the question as to what extent it is possible to inhibit zygomycetes just not only by direct contact with allicin, but also via the gas phase. The susceptibility of both fungi was first compared in an agar diffusion test (Figure 3A): An advantage of the inhibition zone tests is that the areas are easy to measure and are thus quantifiable. As in the drop tests, the susceptibility of *M. racemosus* to allicin was greater than that of *R. stolonifer*, although, of course, here again the different growth behaviour must be considered. It could be speculated, for example—but to our knowledge this has not yet been done for fungi—that the rapid growth is associated with high cytoskeletal dynamics. Since various studies have shown that the tubulin and actin cytoskeleton in human cells is one of the central targets for allicin [29,33], it would be conceivable that allicin targets thiols of the fungal cytoskeleton during rapid growth. However, this is only speculative at this point and would require experimental verification.

Subsequently, the antifungal of allicin vapour was tested (Figure 3C). It was shown that allicin vapour inhibits spore germination of both Mucorales species. Unfortunately, only the initial concentration of the allicin solution in the drop can be given since quantification of allicin in the gas phase is difficult. Due to its temperature instability, allicin itself cannot be detected using gas chromatographic measurements [43,44]. Recently, the measurement of allicin in the gas phase using secondary electrospray ionization coupled Orbitrap mass spectrometry (SESI-Orbitrap MS) was successful, but quantification in various experimental setups remains a challenge [45]. For this reason, we used a simplified computational approach to calculate an estimated effective allicin concentration in the gas phase, which allows a comparison of relative effective concentrations for different application methods. This yielded a similar efficacy for allicin vapour treatment and direct ampB application (Table 1), highlighting the therapeutic potential of allicin via the gas phase, especially since ampB is not a volatile substance and cannot be used in this way.

Again, a clear inhibition of both tested fungi was shown via the gas phase. This observation has direct practical relevance: zygomycetes can lead to mycoses in the nasal cavity, especially in immunosuppressed patients, and in particularly severe cases, grow-through into the cranial cavity and infestation of the brain has been observed [46]. According to the Centers for Disease Control and Prevention (CDC), medical treatment is usually by intravenous or oral administration of antifungal drugs such as amphotericin B, posaconazole or isavuconazole (https://www.cdc.gov/fungal/diseases/mucormycosis/treatment.html, accessed on 10 October 2023). Treatment of local infections with systemic antifungals can lead to undesirable side effects, which is why treating respiratory tract infections with an antifungally active agent via the gas phase is potentially very promising. Using a lung model, we have already shown with thermotolerant fungi that deposition of allicin—either in the gas phase or as an aerosol—is present in the model and shows efficacy [23]. Subsequently, it will now be necessary to extend the model for the zygomycetes to include the nasopharynx and to conduct analogous experiments.

However, there are some challenges associated with the use of allicin as a therapeutic agent to combat respiratory tract infections. On the one hand, allicin can be prepared in large quantities due to its simple synthesis process [47]; on the other hand, the stability of larger amounts of allicin in solution is challenging due to its sensitivity to higher temperatures [48] and dose-dependent damage to human epithelial cells has also been demonstrated (Figure 8). Improper use of garlic preparations has resulted in documented tissue burns [49], which is why treatment with allicin-containing aerosols could pose greater damage to lung tissue than allicin vapour. Suitable allicin concentrations for vapour applications must be determined. Notably, historical reports such as Minchin’s successful treatment of pulmonary tuberculosis with garlic vapor [50], underline the potential of allicin in the treatment of lung diseases.

In order to accurately quantify inhibitory concentrations, EC_50_ values were tested for both fungi considered for allicin, but also for the classical antifungal agent ampB as a reference (Figure 4). Subsequently, the synergistic effect of allicin with ampB was investigated (Figure 7). In summary, three conclusions can be drawn from these Figures: firstly, allicin is already effective against fungi at much lower concentrations than ampB on a mol for mol basis; secondly, in both situations *Rhizopus* is significantly more sensitive than the *Mucor* species to both substances; and thirdly, and perhaps most importantly, allicin and ampB were demonstrated to work together synergistically. This is of direct practical importance and leads to the question, whether an application of allicin (or corresponding substances derived from it) via the gas phase could support a therapy against zygomycete mycoses by means of ampB. Because of the high hazard of these infections, also seen in the context of secondary infections after COVID-19 disease [12,51], this could be a supportive aid for patients who are otherwise at risk. AmpB itself, after all, is known to have significant side effects, for example on renal physiology [52,53], so reducing the pharmacologically necessary concentration of ampB to achieve control of the pathogen is of great importance, also to protect, possibly, multimorbid patients.

Allicin and ampB differ primarily in their mode of action. Whereas ampB interacts with fungal ergosterols to form pores in the membrane and extramembranous sponge-like structures [54,55,56], allicin, as already mentioned, mainly acts as a redox toxin [57], although interactions with the bacterial cell wall and fungal membrane have also been described [58,59]. Another mode of microbial growth inhibition is the disruption of lipid biosynthesis, as has been demonstrated for various *Candida* species [60,61], and the impairment of the expression of genes coding for enzymes of amino acid metabolism, protein degradation, iron acquisition, respiratory chain and thiamine metabolism, as demonstrated in *Sacharomyces cerevisiae* [62]. Allicin could also affect sterol metabolism, because it has been shown that squalene monooxygenase in human cells is inhibited by allicin, and this is a key enzyme for sterol metabolism [63]. As stated earlier, allicin has a variety of potential targets, but a key cellular metabolite involved in protection and resistance against allicin is glutathione [21]. We showed that allicin causes glutathione oxidation, either forming dimeric oxidized glutathione (GSSG) [57] or an allicin-glutathione adduct (*S*-allylmercaptoglutathione, GSSA), which can be reduced back to GSH, at least by yeast glutathione reductase [64]. In different systems we could show that either an increase of the glutathione level leads to an increase of the tolerance to glutathione, but on the other hand also mutations affecting the glutathione metabolism cause an increased sensitivity [21]. Therefore, the absolute glutathione level and the activity of glutathione reductase, which recycles either GSSG or GSSA back to GSH, is crucial for understanding susceptibility and tolerance to allicin. Accordingly, we measured these factors for both fungi.

Contrary to our expectations, *R. stolonifer* was found to have higher levels of both GR activity and absolute glutathione level than *M. racemosus* (Figure 6), which is the inverse of their sensitivity to allicin solutions (Figure 4). This suggests that a different, so far not understood mechanism of tolerance to allicin is present here with respect to that which we have observed in other organisms. Interestingly, it can be observed that the amount of GSSG is relatively high in untreated *R. stolonifer*, although the activity of GR, compared to *M. racemosus*, was higher. Whether this is due to a general increase in tolerance to oxidative stress cannot be determined without further investigation but could explain the observed data to some extent. In bacteria, we identified several other resistance mechanisms to allicin that are largely independent of glutathione [21]. It is possible that such mechanisms also exist in fungi and are not yet understood.

Since allicin attacks the cells via an oxidative mechanism, the different sensitivity of epithelial cells and pathogens to oxidative stress in the respiratory tract is an important factor that needs to be investigated for any inhalation therapy. It must also be taken into account that cells cultured in vitro could be more sensitive than those in vivo, where a continuous supply of GSH could counter depletion through allicin oxidation, and the restoration of GSH in body fluids would have a protective effect on the epithelial cells against inhaled allicin. For example, lung epithelial cells are constantly supplied with blood and epithelial lining fluid containing GSH [65,66], which protects the cells from oxidative stress.

## 4. Materials and Methods

### 4.1. Allicin Synthesis

Allicin synthesis was conducted with slight modifications following the method outlined by Albrecht et al. [47]. Chromatographic separation was replaced with hexane and dichloromethane extraction.

### 4.2. Fungi Procurement and Cultivation Methods

Mucorales fungi *Rhizopus stolonifer* (DSM 855) and *Mucor racemosus* (DSM 5266) were purchased from German Collection of Microorganisms and Cell Cultures GmbH (DSMZ, Braunschweig, Germany).

*R. stolonifer* was routinely cultivated at 22 °C room temperature on PDA plates for two days while *M. racemosus* was cultured at the same conditions for three days.

### 4.3. Growth Assay and Macroscopic Characterization

Fungal growth was studied on different culture media. Pure cultures of *Rhizopus stolonifer* and *Mucor racemosus* were subcultured on five differential media: Czapek agar (CZA), malt extract agar (MEA), potato dextrose agar (PDA), Sabouraud glucose agar (SGA) and oatmeal agar (OA). Cultures were incubated at 22 °C for 40 h for diameter measurements and morphological characterization.

The media were all prepared with distilled water and subsequently sterilized by autoclaving at 121 °C and 1 bar for 30 min. The following quantities refer to a final volume of 1 L. For solid media, if not described otherwise, 1.5% agar (*w*/*v*) was added.

Potato dextrose agar (PDA): 39 g potato glucose agar (Carl Roth GmbH + Co. KG, Karlsruhe, Germany);Czapek agar (CZA): 30 g saccharose, 2.5 g NaNO_3_, 1 g K_2_HPO_4_, 0.5 g KCl, 0.5 g MgSO_4_ · 7H_2_O, 0.01 g FeSO_4_ · 7H_2_O;Malt extract agar (MEA): 15 g malt extract (Carl Roth GmbH + Co. KG, Karlsruhe, Germany);Sabouraud glucose agar (SGA): 40 g glucose monohydrate, 10 g peptone;Oatmeal agar (OA): 30 g of oat flakes were brought to a boil in 1 L deionised water and allowed to simmer gently for 2 h. The oat flakes were then suspended. The suspension was filtered and agar was added to the filtrate.

### 4.4. Inhibition Zone Tests

Spore suspensions from a two (*R. stolonifer*) or three (*M. racemosus*) days old pure plate culture were prepared by flooding the plate with water and transferring 10 mL into a sterile centrifuge tube. The suspension was filled up with distilled water to 50 mL and spore count of 10 µL suspension was determined using a Thoma counting chamber. A sufficient amount of spore suspension was then added to 20 mL molten PDA medium at 50 °C to obtain a spore concentration of 1 × 10^5^ spores per mL. The mixture was poured immediately into a Petri dish to make spore-seeded agar plates. For agar diffusion test, wells (Ø = 0.6 cm) were punched out of the solidified agar with a cork borer. 40 µL solution was added to each well, test solutions were 10 mM and 20 mM aqueous allicin solution and water was used as control. To test antifungal activity of allicin vapor, 25 µL of 25 mM aqueous allicin solution as well as different amounts of 50 mM allicin solution (25 µL and 50 µL) were pipetted onto the Petri dish lid and the solidified agar plate with fungal spores was placed inverted over the lid. 25 µL water and 96% ethanol were used as controls. Growth of *R. stolonifer* was evaluated after 24 h and that of *M. racemosus* after 48 h.

### 4.5. Drop Test

Allicin-containing PDA plates were prepared by adding aqueous allicin solution to 50 mL of molten PDA medium (50 °C). Triplicates were prepared for allicin concentrations of 5 μM, 10 μM, 20 µM and 40 μM, as well as H_2_O_dd_. Serial dilutions of spore suspensions (see Section 4.4; initial concentration: 5 × 10^5^ spores per mL) were prepared. From each dilution, 20 μL was spot pipetted onto the medium. The plates were incubated at 22 °C and fungal growth documented photographically after 24 h.

### 4.6. Determination of the Half Maximal Effective Concentration (EC_50_) and the Minimum Inhibitory Concentration (MIC)

Serial dilutions of allicin in water (0.32–325 µg/mL = 1.95–2000 µM) and amphotericin B (ampB) in 1% DMSO (0.45–462 µg/mL = 0.49–500 µM) were prepared. Spore suspensions were prepared as described in 4.4. 50 µL of a test solution was mixed with 50 µL of spore suspension (final concentration: 2 × 10^3^ spores per mL) in a well of a 96 well microtiter plate. Plates were covered with air-permeable, self-adhesive cling film (Carl Roth GmbH, Karlsruhe, Germany) and incubated without shaking. Spore germination of *R. stolonifer* was studied microscopically after 24 h, that of *M. racemosus* after 48 h. The ratio of germinated spores to total spore number was determined. EC_50_ indicates the concentration at which 50% of the spores did not germinate. The minimum inhibitory concentration (MIC) is defined as the lowest concentration of an antifungal substance that completely inhibits spore germination (≥99%). A low MIC value consequently correlates with high antifungal activity, since small amounts of the antifungal agent are needed to inhibit spore germination.

### 4.7. Investigation of Synergistic Effect of Allicin and AmpB

Serial dilutions of allicin in water (0.19–20 µg/mL = 1.2–125 µM) and ampB in 1% DMSO (3.70–462 µg/mL = 4–500 µM) were prepared and mixed in a 96 well microtiter plate. Spore suspensions were prepared as described in Section 4.4. 50 µL of spore suspension (final concentration: 2 × 10^3^ spores per mL) was added to the 96 well microtiter plate. Plates were covered with air-permeable, self-adhesive cling film (Carl Roth GmbH, Karlsruhe, Germany) and incubated without shaking. Spore germination of *R. stolonifer* was studied microscopically after 24 h and that of *M. racemosus* after 48 h. The ratio of germinated spores to total spore number was determined. 

### 4.8. Calibration of the Glutathione Reductase (GR) and Glutathione Determination Assay

The concept for measuring glutathione reductase (GR) activity originates from a GR recycling assay by Griffith [67], modified by Anderson [68]. In the presence of GR, monomeric glutathione (GSH) was sequentially oxidized to glutathione disulfide (GSSG) by 5,5′-dithiobis-(2-nitrobenzoic acid) (DTNB) consuming NADPH. 2-nitro-5-thiobenzoic acid is formed and its formation can be photometrically detected by changes in absorbance at 412 nm.

The assay was calibrated with a commercially purchased glutathione reductase from *Saccharomyces cerevisiae* (Sigma-Aldrich Chemie GmbH, Taufkirchen, Germany) and various GSSG concentrations (AppliChem GmbH, Darmstadt, Germany). The GR enzyme solution (20 U per mL) was prepared in 143 mM sodium phosphate buffer (pH 7.5) containing 6.3 mM Na_2_EDTA. The following reaction mixture was used: 333 µL H_2_O_dd_, 350 µL NADPH (0.3 mM), 50 µL DTNB (6 mM; Carl Roth, Karlsruhe, Germany), 5 µL yeast GR, 12.5 µL GSSG.

The rate of 2-nitro-5-thiobenzoic acid formation was monitored at 412 nm three times. The concentration was then mapped as a function of the conversion rate and the slopes of the resulting lines were plotted as a function of the GSSG concentrations used, resulting in a calibration line.

### 4.9. Preparation of Cell Lysates

Fungal cell lysates were prepared with slight modifications following the procedure outlined by Gruhlke et al. [57]. Fungal mycelium was scraped from a two- to three-day old plate pure culture with a sterile scalpel and transferred to a 50 mL reaction tube. 3 mL of 143 mM sodium phosphate buffer (pH 7.5) containing 6.3 mM Na_2_EDTA and 500 µL glass beads each (Ø = 1 mm and 4 mm) were added. Samples were vortexed for 1 min and afterwards placed on ice for another minute. This step was repeated six times. Subsequently, cell debris was pelleted by centrifugation (15 min, 3000× *g*, 22 °C). The supernatant containing the cell lysate was transferred to a 2 mL reaction tube and used to measure glutathione reductase activity and cellular glutathione levels.

### 4.10. Determination of Protein Levels by Bradford Assay

To determine the amount of protein in a sample, the photometric method according to Bradford was used [69]. Bradford solution was prepared by dissolving 10 mg Coomassie Brilliant Blue G-250 (Carl Roth, Karlsruhe, Germany) in 5 mL ethanol (96%). 11 mL of phosphoric acid (85%, Carl Roth, Karlsruhe, Germany) was added and the solution was made up to 100 mL with H_2_O_dd_. The assay was calibrated with a serial dilution of BSA (2 mg/mL). For protein levels determination, 900 μL Bradford reagent was mixed with 100 μL cell lysate and the absorbance of the sample was measured photometrically at 595 nm (P4 UV/Vis spectrophotometer, VWR, Radnor, PA, USA).

### 4.11. Glutathione Reductase Enzyme Activity Assay

For the measurement of glutathione reductase activity, the previously prepared fungal cell lysate was used instead of commercial enzyme. The assay was performed according to Griffith et al. [67] (modified by [68]). Prior to the assay, substrate concentration ranges giving linear kinetics were established at enzyme excess ensuring the rate of substrate conversion was proportional to the substrate concentration. The reaction mixture composition for measuring fungal GR activity was as follows: 333 µL H_2_O_dd_, 350 µL NADPH (0.3 mM), 50 µL DTNB (6 mM), 20 µL fungal cell lysate, 12.5 µL GSSG (1.25 mM, 1.0 mM, 0.75 mM, 0.5 mM, 0.25 mM). The rate of 2-nitro-5-thiobenzoic acid formation was monitored photometrically at 412 nm. Cell lysates of three biological replicates with three technical replicates each were measured for each fungus. GR activity was standardized to protein content in cell lysate.

### 4.12. Determination of Cellular Glutathione Levels

Glutathione levels were determined using the enzymatic recycling assay based on the GR. The assay was performed according to Griffith et al. [67] (modified by [68]). Fungal cell lysate (see above) was used for tests. Total glutathione was read from the standard curve. Adding 2-vinylpyridine removes GSH from the cell lysate, so the assay reflects only the GSSG concentration.

### 4.13. Allicin Toxicity on Human Epithelial Cell Lines

Allicin toxicity to the non-small cell lung cancer cell line A549 (CCL-185) from human adenocarcinoma alveolar epithelial cells and the WISH (CCL-25) cell line (origin: human papillomavirus-related cervical adenocarcinoma) was examined. A549 cells were routinely cultivated in RPMI-1640 and WISH cells in DMEM (both Sigma-Aldrich, Steinheim, Germany). The culture media were supplemented with 10% FCS (Bio&SELL GmbH, Nürnberg, Germany) and 2 mM L-glutamine, 100 U/mL potassium penicillin, and 100 µg/mL streptomycin sulfate (all from Sigma-Aldrich).

Cells were incubated with 100 µL of allicin-containing medium (20 µM–2500 µM) for 1 h at 37 °C with 5% CO_2_. The medium was then replaced with normal culture medium and 20 µL MTT solution (5 mg/mL) was added. The MTT-containing medium was removed after 1.5–3 h of incubation and the formazan was dissolved with dimethyl sulfoxide. The absorbance at 570 nm (reference wavelength: 690 nm) was measured using the SpectraMax i3x Multi-Mode microplate reader (Molecular Devices, San José, CA, USA). EC_50_ indicates the concentration at which cell viability was reduced by 50%.

### 4.14. Statistical Analysis

Statistical analyses were performed using SigmaStat, Version 3.11.0 (Systat Software, San José, CA, USA). A *p* value ≤ 0.05 was considered statistically significant. Statistical significance of inhibition zone tests was calculated by One-Way ANOVA using the Holm–Šidák method. Same letters indicate no significant difference. For the comparison of two variables, a Student’s *t*-test was used to calculate statistical significance. Significant differences were marked by asterisks (*p* < 0.001 = ***).

## 5. Conclusions

Overall, our study confirms the sensitivity of zygomycetes, which are a major cause of mycoses, to the natural substance allicin. Both direct contact with allicin solutions and gas phase exposure demonstrated allicin’s efficacy, which was comparable to the classical antifungal agent ampB. Since a synergistic effect with ampB was shown, this could be a first step towards a possible application of allicin or derived substances via the gas phase, for the treatment of either isolates highly tolerant to ampB or for the reduction of the necessary concentration of ampB to contain the infection and thus protect patients. Despite promising results, challenges such as allicin’s instability and potential damage to human epithelial cells must be addressed. The unexpected inverse relationship between glutathione levels and allicin sensitivity in *R. stolonifer* warrants further investigation into tolerance mechanisms. Future research should focus on optimizing allicin concentrations, understanding synergistic effects with ampB, and exploring its therapeutic potential in respiratory infections.

## Figures and Tables

**Figure 1 ijms-24-17519-f001:**
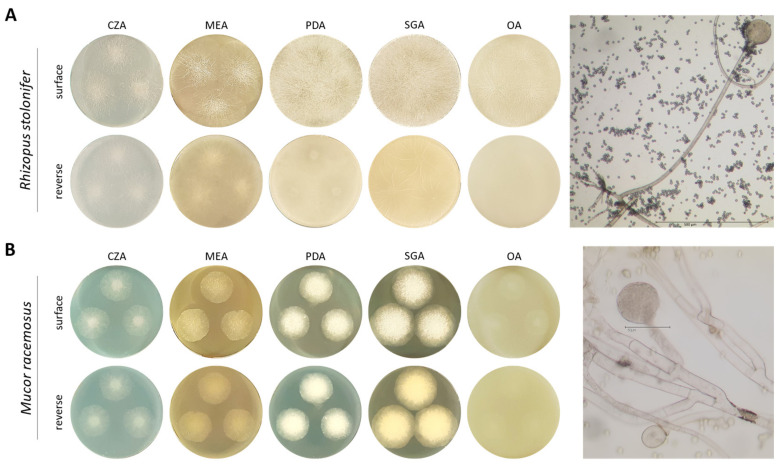
Macroscopic and microscopic growth characteristics of (**A**) *Rhizopus stolonifer* and (**B**) *Mucor racemosus*. Growth was studied on Czapek agar (CZA), malt extract agar (MEA), potato dextrose agar (PDA), Sabouraud glucose agar (SGA) and oatmeal agar (OA). Spore suspension (20 µL 5 × 10^5^ spores per mL) was pipetted three times onto different solid media and incubated at 22 °C for 40 h. Surface growth and colony reverse were documented photographically. Fungal reproductive structures were examined using transmitted light microscopy (light microscope DMRBE, Leica, Wetzlar, Germany).

**Figure 2 ijms-24-17519-f002:**
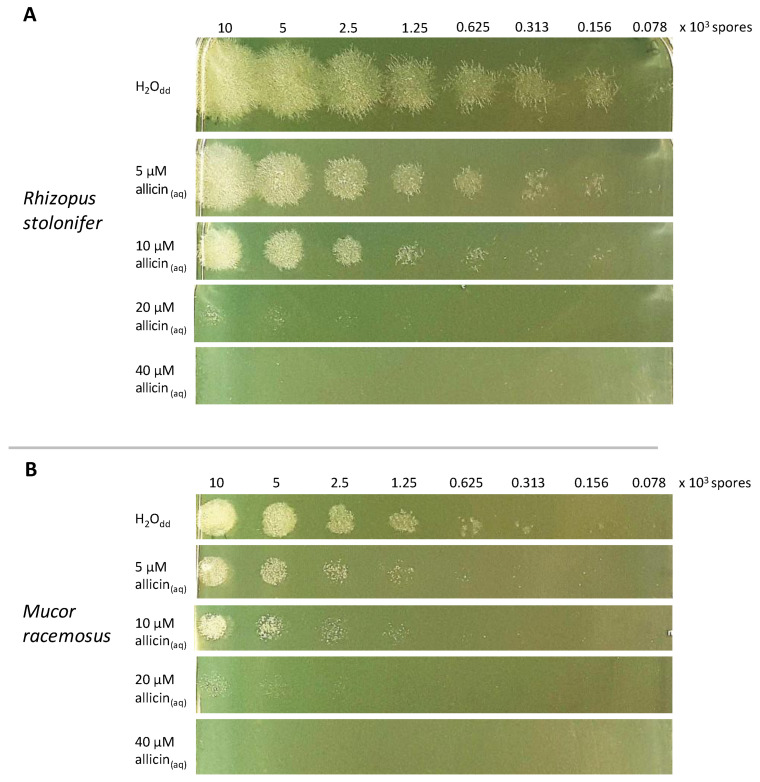
Drop test showing allicin sensitivity of Mucorales strains. 20 µL of serial diluted spore suspensions of (**A**) *Rhizopus stolonifer* and (**B**) *Mucor racemosus* (stock: 5 × 10^5^ spores per mL) were plated onto PDA medium containing allicin at the concentrations indicated. Double distilled water (H_2_O_dd_) was used as a control. Growth was documented after 24 h of incubation at 22 °C. For a better visualisation of growth, the contrast of the photos was increased.

**Figure 3 ijms-24-17519-f003:**
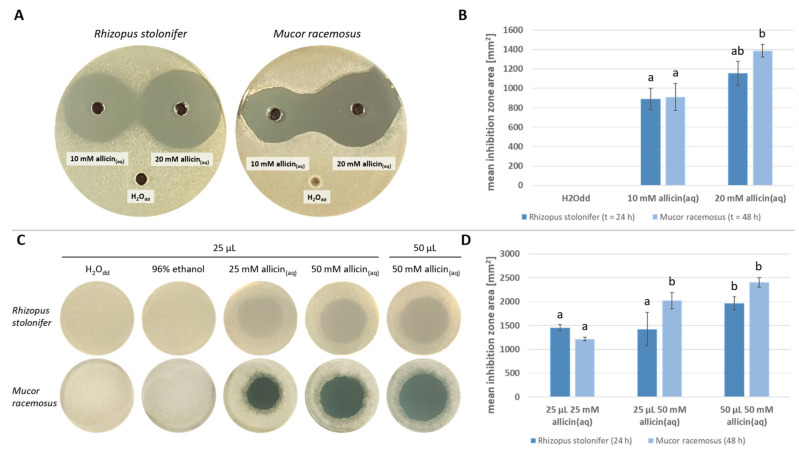
Allicin inhibits spore germination in a concentration-dependent manner. Spore containing agar was prepared by adding spore suspension to 20 mL PDA medium (50 °C) to obtain a final concentration of 10^5^ spores per mL. Triplicates were made for each fungus. *R. stolonifer* was incubated for 24 h and *M. racemosus* for 48 h at 22 °C. (**A**) For agar diffusion tests, three holes (Ø = 0.6 cm) were punched into the agar after solidification and filled with 40 µL of 10 mM, 20 mM aqueous allicin solution or H_2_O_dd_, respectively. (**B**) *n* = 3, error bars show standard deviation. Same letters indicate no significant difference (*p* > 0.05) in a One-Way ANOVA with Holm–Šidák method. (**C**) For vapour treatments, H_2_O_dd_, 96% ethanol or aqueous allicin solution (25 mM resp. 50 mM) was applied to the Petri dish lid after agar solidification. (**D**) *n* = 3, error bars show standard deviation. Same letters indicate no significant difference (*p* > 0.05) in a One-Way ANOVA with Holm–Šidák method.

**Figure 4 ijms-24-17519-f004:**
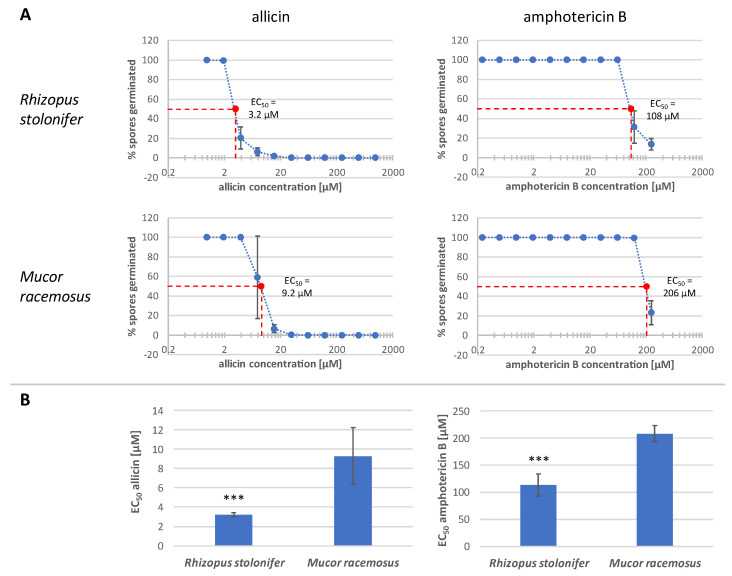
Allicin inhibits spore germination more effectively than amphotericin B. (**A**) Fungal spore suspension was mixed with sample solution. Spore germination of *R. stolonifer* was studied microscopically after 24 h, that of *M. racemosus* after 48 h. The ratio of germinated spores to total spore number was determined (H_2_O_dd_ or 1% DMSO was set to 100% spore germination). The mean values of three biological replicates with three technical replicates each are shown. Sample solution concentration was plotted logarithmically. Error bars represent the standard deviation. (**B**) *n* = 9, sample comprises three biological replicates with three technical replicas each, error bars show standard deviation. Significant differences between species in a Student’s *t*-test are marked by asterisks (*p* < 0.001 = ***).

**Figure 5 ijms-24-17519-f005:**
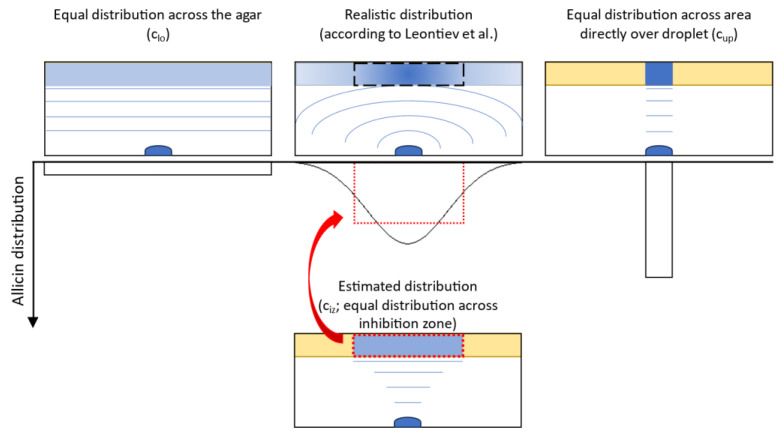
Visualization of different models for the deposition of allicin via the gas phase. The realistic distribution in an inverted inhibition zone test according to [38] (**top center**) was simplified by assuming an equal allicin distribution across the inhibition zone (c_iz_, **bottom center**). Equal distribution across the entire agar (**left**) and closely confined distribution directly over the droplet (**right**) correspond to the boundary concentrations c_lo_ and c_up_ in the calculations.

**Figure 6 ijms-24-17519-f006:**
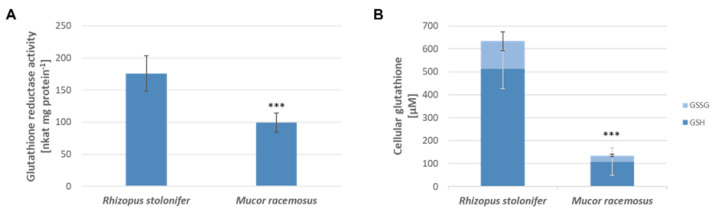
Comparison of GR activity and cellular glutathione content of *R. stolonifer* and *M. racemosus.* (**A**) Mean glutathione reductase activity and (**B**) cellular glutathione in fungal cell lysates were measured in a glutathione reductase recycling assay (Section 4.11 and Section 4.12). Protein content in cell lysates was determined using a Bradford assay (Section 4.10). GR activity was standardized to protein content. *n* = 9, sample comprises three biological replicates with three technical replicas each, error bars show standard deviation. Significant differences in a Student’s *t*-test are marked by asterisks (*p* < 0.001 = ***).

**Figure 7 ijms-24-17519-f007:**
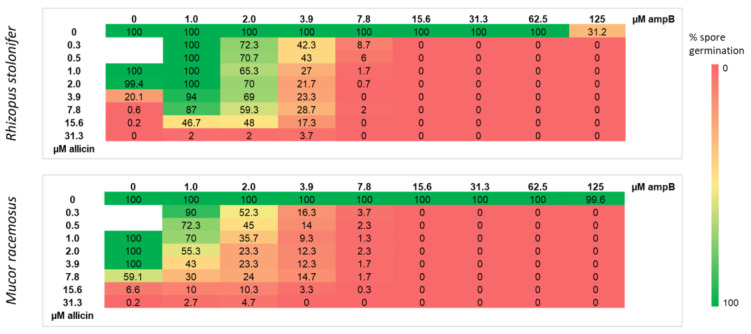
Allicin enhances the antifungal activity of amphotericin B. Allicin was placed in a 96 well plate along with ampB and 50 µL spore suspension was added. Final concentrations of the sample solutions per well are given. Spore germination of *R. stolonifer* was studied microscopically after 24 h, that of *M. racemosus* after 48 h. The ratio of germinated spores to total spore number was determined (Spore germination without antifungal agent was set as 100% spore germination. Mean values of three biological replicates are shown. Inhibition of spore germination is represented by color: red—inhibition, green—no inhibition.

**Figure 8 ijms-24-17519-f008:**
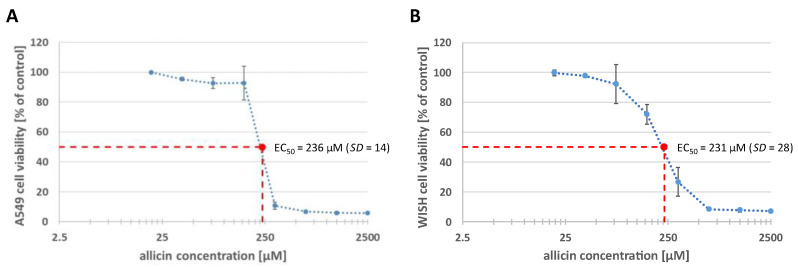
Dose-dependent toxicity of allicin on human epithelial cell lines. A549 (**A**) and WISH (**B**) cell lines were incubated with 100 µL allicin-containing media (20 µM–2500 µM) for 1 h. The medium was changed and 20 µL MTT (5 mg/mL) was added. Cells were incubated at 37 °C with 5% CO_2_. MTT-containing medium was removed and formazan was dissolved with dimethyl sulfoxide (DMSO). The absorbance at 570 nm (reference: 690 nm) was measured (SpectraMax i3x Multi-Mode microplate reader, Molecular Devices, San José, CA, USA). *n* = 3–4, mean values are shown, error bars show standard deviation.

**Table 1 ijms-24-17519-t001:** Summary of the mean effective concentration of the application types for inhibiting spore germination of Mucorales.

Mucorales Species	Mean Effective Concentration of Application Types to Inhibit Spore Germination [µM]			
	Allicin			Amphotericin B
	Vapour	Diffusion	Direct Contact	Direct Contact
*Rhizopus stolonifer*	255	142	31.3	>250
*Mucor racemosus*	210	118	31.3	>250

## Data Availability

The data that support the findings of this study are available from the first author upon reasonable request.

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
