# Peer review of "Combating Black Fungus: Using Allicin as a Potent Antifungal Agent against Mucorales"

_ijms, 2023, doi:10.3390/ijms242417519_

Round 1

Reviewer 1 Report

Comments and Suggestions for Authors

Manuscript ID: ijms-2733218
"
 Combating black fungus: allicin as a potent antifungal agent against Mucorales"

The paper provides insights into a bioactive compound (allicin) as an antifungal agent and its potentiality in combating black fungus. The paper is well-written and has clinical applications.

I suggest the following minor revisions before considering this manuscript again for publication.

Please find below the comments:

Ø  Restructure the abstract highlighting the methods and potential values, avoiding descriptive abstract.

Ø  The authors are suggested to add in-depth discussion and compare with existing literature about the probable mechanism of action of allicin.

Ø  Authors are suggested to add other bioactive compounds information potent against black fungus in the introduction section.    

Comments on the Quality of English Language

The manuscript is well written but in the results and discussion sections still more clarity is required. 

Author Response

Point-by-point reply to reviewer 1

The paper provides insights into a bioactive compound (allicin) as an antifungal agent and its potentiality in combating black fungus. The paper is well-written and has clinical applications.

We thank the reviewer for its overall positive evaluation.

I suggest the following minor revisions before considering this manuscript again for publication.

Please find below the comments:

Ø  Restructure the abstract highlighting the methods and potential values, avoiding descriptive abstract.

We thank the reviewer for this helpful comment and included methods and potential values in lines 20-26 of the abstract.

Ø  The authors are suggested to add in-depth discussion and compare with existing literature about the probable mechanism of action of allicin.

We were happy to include further information in lines 476-480 of the discussion.

Ø  Authors are suggested to add other bioactive compounds information potent against black fungus in the introduction section.    

We included other bioactive compounds in lines 73-80 of the introduction as requested.

Comments on the Quality of English Language

The manuscript is well written but in the results and discussion sections still more clarity is required. 

A native speaker checked the manuscript.

Reviewer 2 Report

Comments and Suggestions for Authors

-Unfortunately, in the Introduction section, the authors did not share and compare the literature findings on the antifungal effect of allicin. The advantages and innovation aspects of this research compared to literature studies need to be explained in detail. More data is needed.

-Allicin is an antiviral, antifungal, antioxidant and antibacterial bioactive compound. These features should be emphasized more clearly in the introduction section. Research conducted in this field will give ideas to authors.

Encapsulation and antibacterial studies of goji berry and garlic extract in the biodegradable chitosan

G Baysal, HS Olcay, Ç Günneç

Journal of Bioactive and Compatible Polymers 38 (3), 209-219

-n case of R. stolonifer and 2018 mm2

(SD = 167) for M. racemosus. Application of 161

50 µL 50 mM allicin resulted the largest inhibition zones for both fungi. Here, the mean 162

inhibition zone area of R. stolonifer was 1965 mm2

(SD = 137) and 2405 mm2

(SD = 100) for.....the highest inhibition zone was obtained against M. racemosus. Please cite the reason for this in detail.

-Please provide literature references to the equations and equations used in the manuscript.

-There are almost no literature references for experimental processes under titles such as 4.9, 4.10, 4.11, 4.12. Please add.

-In the discussion section, add a table comparing the analysis findings with the literature findings to compare the results.

-

Comments on the Quality of English Language

Minor editing of English language required

Author Response

Point-by-point reply to reviewer 2

-Unfortunately, in the Introduction section, the authors did not share and compare the literature findings on the antifungal effect of allicin. The advantages and innovation aspects of this research compared to literature studies need to be explained in detail. More data is needed.

We acknowledge the comment of reviewer and added some information in lines 86-89 of the introduction. However, most comparative information regarding allicin are included in the discussion and we wanted to avoid repetitive information in both sections.

-Allicin is an antiviral, antifungal, antioxidant and antibacterial bioactive compound. These features should be emphasized more clearly in the introduction section. Research conducted in this field will give ideas to authors.

Encapsulation and antibacterial studies of goji berry and garlic extract in the biodegradable chitosan

G Baysal, HS Olcay, Ç Günneç

Journal of Bioactive and Compatible Polymers 38 (3), 209-219

We thank the reviewer for this interesting reference and included it in the new lines 86-89 of the introduction.

-n case of R. stolonifer and 2018 mm2

(SD = 167) for M. racemosus. Application of 161

50 µL 50 mM allicin resulted the largest inhibition zones for both fungi. Here, the mean 162

inhibition zone area of R. stolonifer was 1965 mm2

(SD = 137) and 2405 mm2

(SD = 100) for.....the highest inhibition zone was obtained against M. racemosus. Please cite the reason for this in detail.

The difference between the inhibition of the two fungies in the highest allicin concentration is not significant. Therefore, the size of the inhibition area seems to be randomly a little bit larger. We just described the results, but due to the missing significance an interpretation of the difference would not be valuable.

-Please provide literature references to the equations and equations used in the manuscript.

As requested, we included the new reference 38 and for more clarity we draw the cartoon in figure 5 to make the equation visible.

-There are almost no literature references for experimental processes under titles such as 4.9, 4.10, 4.11, 4.12. Please add.

Many thanks for this hint, we did not wanted to repetitively cite the same reference. However, now we included the respective references as requested.

-In the discussion section, add a table comparing the analysis findings with the literature findings to compare the results.

We acknowledge the comment and extended the discussion on this point in lines 476-480. However, we did not include a table due to missing comparability in the different experimental system for detecting the inhibition constants. As you can see in our different models in figures 3 and 4. This would produce more confusion for the readers.

Comments on the Quality of English Language

Minor editing of English language required

A native speaker checked the manuscript.

Reviewer 3 Report

Comments and Suggestions for Authors

The presented article deals with an interesting and important issue. I have several comments and questions about this work.

The abstract provides a general overview of the researched issue, the authors could try to incorporate into it the most important results supplemented with specific data found.

Keywords - words that are in the title of the article should be avoided and it would be advisable to arrange them alphabetically.

Introduction – provides a good introductory overview of the researched issue. Of the 31 cited publications, 12 are self-citations, which seems to me to be a relatively large number. It would be appropriate to more clearly point to other theoretical starting points of the given issue and publications by other authors.

Results – clearly processed issue of own results.

In Discussion - the results of the authors of this article are confronted with previous publications. Consider whether the new results are sufficiently clarified and shown to be beneficial compared to older articles by these authors. This chapter is quite extensive and quotes heavily from their own previous publications. What is the cause?

I am missing Conclusions in the article. I think it would be interesting to increase the attractiveness of the article for readers to include Conclusions in which the achieved results would be summarized. The starting point could be the last paragraph of the Discussion section, from lines 441 to 447. However, only the actual new results should be summarized, avoiding unsubstantiated speculations and citations.

References – a total of 19 authors' own publications from 52 references are cited in the article. I recommend considering whether all self-citations are really necessary to cite for this article.

Author Response

Point-by-point reply to reviewer 3

The presented article deals with an interesting and important issue. I have several comments and questions about this work.

We thank the reviewer for its overall positive evaluation.

The abstract provides a general overview of the researched issue, the authors could try to incorporate into it the most important results supplemented with specific data found.

We thank the reviewer for this hint and included results in the new abstract in lines 20-26.

Keywords - words that are in the title of the article should be avoided and it would be advisable to arrange them alphabetically.

We acknowledge the reviewer comment and arranged the keyword alphabetically. However, allicin is the only keyword which is in the title as well. Since allicin is the major topic, we think this should be in the keywords, but could be deleted if this is the journals style.

Introduction – provides a good introductory overview of the researched issue. Of the 31 cited publications, 12 are self-citations, which seems to me to be a relatively large number. It would be appropriate to more clearly point to other theoretical starting points of the given issue and publications by other authors.

The reviewer is completely right and we reduced the number of self-citations and included additional references from other authors. In all we end up now at 10 self-citations of 69 references, which are 14.5% instead of 36,5% (19 of 52) before. This is now less than 15% as requested by MDPI. Sorry, that we haven´t seen this before.

Results – clearly processed issue of own results.

Many thanks for this comment.

In Discussion - the results of the authors of this article are confronted with previous publications. Consider whether the new results are sufficiently clarified and shown to be beneficial compared to older articles by these authors. This chapter is quite extensive and quotes heavily from their own previous publications. What is the cause?

We reduced self-citations as indicated above.

I am missing Conclusions in the article. I think it would be interesting to increase the attractiveness of the article for readers to include Conclusions in which the achieved results would be summarized. The starting point could be the last paragraph of the Discussion section, from lines 441 to 447. However, only the actual new results should be summarized, avoiding unsubstantiated speculations and citations.

Many thanks for this comment. We no included a conclusion.

References – a total of 19 authors' own publications from 52 references are cited in the article. I recommend considering whether all self-citations are really necessary to cite for this article.

Self-citations are reduced as indicated above.

Reviewer 4 Report

Comments and Suggestions for Authors

The manuscript entitled “Combating black fungus: allicin as a potent antifungal agent against Mucorale” written by Schier and coworkers discusses the antifungal assessment of allicin in both contact and gas phases as well as drawing its comparisons to amphotericin B. I find the study interesting and if the editorial team finds it aligns with the scientific focus and level of IJMS, I endorse its publication. I would only recommend incorporating an evaluation of allicin's toxicity against human cell lines, or at the very least, discussing it in the context of IC50 values obtained previously in literature reports.

Comments on the Quality of English Language

Minor editing of language required

Author Response

Point-by-point reply to reviewer 4

The manuscript entitled “Combating black fungus: allicin as a potent antifungal agent against Mucorale” written by Schier and coworkers discusses the antifungal assessment of allicin in both contact and gas phases as well as drawing its comparisons to amphotericin B. I find the study interesting and if the editorial team finds it aligns with the scientific focus and level of IJMS, I endorse its publication. I would only recommend incorporating an evaluation of allicin's toxicity against human cell lines, or at the very least, discussing it in the context of IC50 values obtained previously in literature reports.

We completely agree to the reviewer comments and added the EC50 of epithelial cells in new figure 8 as requested. We think EC50 is more convenient for living cells than IC50.

Comments on the Quality of English Language

Minor editing of language required

A native speaker checked the manuscript.

Reviewer 5 Report

Comments and Suggestions for Authors

The manuscript presents a biological exploration of allicin as a potential agent against zygomycetes through direct contact and gas phase exposure. It comes as a continuation of prior research conducted by the same group in this field.

This research aligns with a current area of interest, as recent studies have explored similar applications with various fungi. The findings are intriguing and contribute a valuable piece to the puzzle in the ongoing research aimed at comprehensively understanding the mechanism of action of allicin and its potential applications.

The manuscript is well written and accessible to the readers; the results are clearly presented and interpreted.

I recommend introducing a small discussion relation to the potential toxicity of allicin when used in gas phase (if the necessary concentration to use is safe for a potential application).

Author Response

Point-by-point reply to reviewer 5

The manuscript presents a biological exploration of allicin as a potential agent against zygomycetes through direct contact and gas phase exposure. It comes as a continuation of prior research conducted by the same group in this field.

This research aligns with a current area of interest, as recent studies have explored similar applications with various fungi. The findings are intriguing and contribute a valuable piece to the puzzle in the ongoing research aimed at comprehensively understanding the mechanism of action of allicin and its potential applications.

The manuscript is well written and accessible to the readers; the results are clearly presented and interpreted.

We thank the reviewer for this overall positive comment.

I recommend introducing a small discussion relation to the potential toxicity of allicin when used in gas phase (if the necessary concentration to use is safe for a potential application).

We agree to the reviewer that this would be helpful. We added this in lines 445-455 of the discussion.

Round 2

Reviewer 3 Report

Comments and Suggestions for Authors

The article was partially edited according to the reviewer's instructions and the authors' possibilities. I recommend it for publication.